# Tomato Spotted Wilt Virus Suppresses the Antiviral Response of the Insect Vector, *Frankliniella occidentalis*, by Elevating an Immunosuppressive C18 Oxylipin Level Using Its Virulent Factor, NSs

**DOI:** 10.3390/cells13161377

**Published:** 2024-08-19

**Authors:** Niayesh Shahmohammadi, Falguni Khan, Gahyeon Jin, Minji Kwon, Donghee Lee, Yonggyun Kim

**Affiliations:** 1Department of Plant Medicals, Andong National University, Andong 36729, Republic of Korea; niayeshshahmohammadi@gmail.com (N.S.); falguni.agri@gmail.com (F.K.); gsh07129@daum.net (G.J.); 2Industry Academy Cooperation Foundation, Andong National University, Andong 36729, Republic of Korea; lincplus2@anu.ac.kr (M.K.); dhlee10@anu.ac.kr (D.L.)

**Keywords:** *Orthotospovirus tomatomaculae*, western flower thrips, immune response, apoptosis, EpOME

## Abstract

*Orthotospovirus tomatomaculae* (tomato spotted wilt virus, TSWV) is transmitted by the western flower thrips, *Frankliniella occidentalis*. Epoxyoctadecamonoenoic acids (EpOMEs) function as immune-suppressive factors, particularly in insects infected by viral pathogens. These oxylipins are produced by cytochrome P450 monooxygenases (CYPs) and are degraded by soluble epoxide hydrolase (sEH). In this study, we tested the hypothesis that TSWV modulates the EpOME level in the thrips to suppress antiviral responses and enhance its replication. TSWV infection significantly elevated both 9,10-EpOME and 12,13-EpOME levels. Following TSWV infection, the larvae displayed apoptosis in the midgut along with the upregulated expression of four caspase genes. However, the addition of EpOME to the viral treatment notably reduced apoptosis and downregulated caspase gene expressions, which led to a marked increase in TSWV titers. The *CYP* and *sEH* genes of *F. occidentalis* were identified, and their expression manipulation using RNA interference (RNAi) treatments led to significant alternations in the insect’s immune responses and TSWV viral titers. To ascertain which viral factor influences the host EpOME levels, specialized RNAi treatments targeting genes encoded by TSWV were administered to larvae infected with TSWV. These treatments demonstrated that *NS_S_* expression is pivotal in manipulating the genes involved in EpOME metabolism. These results indicate that NSs of TSWV are crucially linked with the elevation of host insect EpOME levels and play a key role in suppressing the antiviral responses of *F. occidentalis*.

## 1. Introduction

The tomato spotted wilt virus (TSWV), *Orthotospovirus tomatomaculae*, is a negative-sense single-stranded RNA virus classified within the genus *Orthotospovirus* and the family *Tospoviridae* [1]. TSWV becomes a substantial economic threat to global agriculture due to its ability to infect over 1000 plant species spanning more than 90 families [2,3]. Its genome is divided into three segments (small, medium, and large) and encodes five proteins. Three of these proteins—a nucleocapsid (N) gene, a glycoprotein with two domains (Gn/Gc), and an RNA-dependent RNA polymerase (RdRp), which is essential for viral genome replication and transcription—are conserved across all Bunyaviruses [4]. Furthermore, the genome contains two additional genes that are specifically adapted for interactions with plants and insects: the silencing suppressor, NSs, which helps to counteract host antiviral RNA silencing, and the movement protein, NSm, which facilitates both cell-to-cell and long-distance movement within plants [5].

TSWV is primarily transmitted in a circulative-persistent manner by the western flower thrips, *Frankliniella occidentalis*, an efficient vector of the virus [6]. The interaction between TSWV, its vector, and host plants involves complex molecular mechanisms that enhance viral transmission [7,8]. This interaction seems to provide an evolutionary benefit by allowing the virus to manipulate vector behavior and performance, thus increasing the likelihood of its acquisition and spread within various ecosystems [9]. The relationship between thrips and TSWV is unique because adult thrips can only transmit TSWV if they have acquired the virus during their early larval stages [10,11]. In the immature stages, the virus replicates within the midgut and adjacent tissues before infecting the primary salivary glands [12,13]. Consequently, active viral replication in the insect vector is essential for its horizontal transmission to host plants. To enhance its acquisition by insect vectors, TSWV infection modulates the host plants, downregulating insect resistance and increasing nutritional contents to support the survival, development, and reproduction of the insect vectors [14,15,16,17]. Specifically, NSs, expressed in TSWV-infected plants, inhibit the production of monoterpenes that repel *F. occidentalis* by directly interacting with MYCs, key regulators of the jasmonic acid signaling pathway [18]. Therefore, it is suggested that TSWV infection adapts plant physiology to improve vector fitness [19].

On the other hand, recent investigations have elucidated the complex interactions between TSWV and its vector, illustrating the virus’s tactics for circumventing host immune responses to facilitate effective transmission [7]. One proposed mechanism indicates the induction of immune responses in *F. occidentalis* through the eicosanoid signaling pathway following TSWV infection [20]. There are two primary types of insect innate immunity: cellular or hemocytic immunity, which is triggered by the detection of a threat and primarily removes invading microbes from the hemolymph in the initial stages of infection [21], and humoral immunity, which involves activating genes that synthesize antimicrobial peptides (AMPs), which are effective against invading microbes on and within cellular membranes [22]. Eicosanoids, including prostaglandins [PGs], leukotrienes, and epoxyeicosatrienoic acids, compose a family of oxygenated polyunsaturated fatty acids (PUFAs) with 20 carbons that trigger cellular and humoral immune responses [23]. The biosynthesis of eicosanoids is quickly elevated in response to pathogenic infections, enhancing the expression and activity of phospholipase A_2_ (PLA_2_), which liberates arachidonic acid (20:4n-6) and other C20 PUFAs from cellular membrane phospholipids [24,25]. In contrast to the immune-promoting effects of eicosanoids [26], epoxyoctadecamonoenoic acids (EpOMEs), such as vernolic acid (12,13-EpOME) and coronaric acid (9,10-EpOME), serve as immune-suppressive agents [27,28]. They are produced by cytochrome P450 monooxygenases (CYPs) and are broken down by soluble epoxide hydrolase (sEH) [29].

Apoptosis is crucial for numerous biological functions, including the immune response, which culminates in the self-destruction of cells [30]. It is also key in defending against viral infections [31]. Research has demonstrated that EpOMEs modulate the extent of immune reactions to viral infections in late stages by curbing antiviral apoptosis within lepidopteran insects [28].

Transcriptome analysis revealed TSWV-associated immune components, including the biosynthesis of AMPs, lectin as a pattern recognition receptor, and a Toll receptor to mediate downstream immune responses [32]. Investigation into TSWV immune responses at the protein level showed the expression of stress-associated proteins such as heat shock protein 70, ubiquitin, and dermcidin [33]. Han and Rotenberg [34] identified substantial expressions of lysozyme, apolipophorin, and trypsin in TSWV responses. Kim et al. [20] demonstrated that TSWV infection induces *F. occidentalis* immune responses, activated by eicosanoids from infection foci in the midguts. Thus, the inhibition of PLA_2_ activity markedly suppressed melanization, apoptosis, and the expression of AMP genes [20]. Consequently, the eicosanoid signal should be suppressed while the EpOME signal might be favored by TSWV in *F. occidentalis* to facilitate viral replication.

This study proposed a hypothesis that TSWV upregulates the EpOME signal, which antagonizes the induced immune responses in *F. occidentalis*. To evaluate this hypothesis, this study monitored the EpOME levels in thrips post-viral infection. To support EpOME synthesis and degradation, *CYP* and *sEH* genes involved in EpOME metabolism were identified. The viral factor(s) derived from TSWV was then pinpointed through individual RNA interference (RNAi) techniques targeting viral genes in the infected thrips.

## 2. Materials and Methods

### 2.1. Insect Rearing

A laboratory population of *F. occidentalis* originated from hot pepper (*Capsicum annuum* L.) fields in Andong, Korea was cultured at a temperature of 25 ± 2 °C, a photoperiod of 16:8 h (L:D), and a relative humidity of 60 ± 5%. Germinated beans (*Phaseolus coccineus* L.) were provided for feeding and oviposition. Newly laid eggs in adult colonies were transferred to a breeding dish (SPL Life Sciences, Pocheon, Republic of Korea). After hatching, fresh beans were supplied daily. Under these laboratory conditions, larvae passed through two instars (L1–L2) before developing into prepupae, distinct from the pupae with visible wing pads.

### 2.2. Chemicals

12,13-EpOME was obtained from Cayman (Ann Arbor, MI, USA). 12-(3-adamantan-1-yl-ureido) dodecanoic acid (AUDA), benzyloxycarbonyl-Val-Ala-Asp-fluoromethyl ketone (Z-VAD-FMK), and 5,8,11,14-eicosatetraenoic acid (AA) were sourced from Sigma Aldrich Korea (Seoul, Republic of Korea) and dissolved in dimethyl sulfoxide (DMSO) for the preparation of testing solutions. 5-Bromo-2′-deoxyuridine (BrdU) and anti-BrdU antibody were procured from Abcam (Cambridge, UK). Terminal deoxynucleotidyl transferase, FITC-conjugated anti-mouse IgG antibody, and 4,6-diamidino-2-phenylindole (DAPI) were acquired from Thermo Fisher Scientific (Wilmington, DE, USA). Bovine serum albumin (BSA), dimethylsulfoxide (DMSO), and t-octylphenoxy-polyethoxyethanol (Triton X-100) were obtained from Sigma Aldrich Korea. Phosphate-buffered saline (PBS) was formulated with 100 mM phosphoric acid containing 0.7% NaCl and adjusted to a pH of 7.4 using 1 M NaOH. An anticoagulant buffer (ACB, pH 4.5) was created using 186 mM NaCl, 17 mM Na_2_EDTA, and 41 mM citric acid.

### 2.3. TSWV Infection to Thrips

TSWV was isolated from hot pepper leaves exhibiting typical viral symptoms, including ring spots, in Andong, Republic of Korea, and verified with an Immunostrip TSWV kit (Agdia, Elkhart, IN, USA). Approximately 100 mg of the plant tissues were homogenized in 1 mL of filter (0.22 µm pore size)-sterilized PBS and centrifuged at 14,000× *g* for 5 min. The supernatant was employed as the virus suspension. The viral infection followed the method described by Kim et al. [20]. Briefly, prior to the immune challenge, L1 or L2 larvae underwent a one-hour starvation period. The viral infection was administered via a feeding method in which sprouted bean seed kernels were immersed in 1 mL of the virus suspension for 5 min and subsequently dried for 10 min under aseptic conditions. The infected kernels were then placed in a small breeding Petri dish (100 × 40 mm, SPL Life Sciences) where the test insects were fed for 12 h. After feeding, the virus-infected kernels were substituted with fresh ones. 

### 2.4. Insect Sample Preparation to Quantify EpOMEs

L1 larvae of *F. occidentalis* were utilized for the extraction of EpOMEs. Following a 24-h diet that was infected with TSWV, 1500 larvae were gathered and rinsed three times with chilled PBS. These sample preparations were independently replicated three times. EpOME extraction followed the method described by Vatanparast et al. [2]. Briefly, each sample underwent three cycles of homogenization (1 min per cycle) in PBS using an ultrasonicator (Bandelin Sonoplus, Berlin, Germany) at 80% power, and the pH was then adjusted to 4 using 1 N HCl. The sample was combined with one mL of ethyl acetate to separate the organic upper phase. The aqueous lower phase was re-extracted twice with ethyl acetate. The collected ethyl acetate extracts were concentrated to approximately 500 μL under a gentle stream of nitrogen and were applied to a small silicic acid column (2 × 90 mm containing 30 mg of Type 60A, 100–200 mesh silicic acid, Sigma Aldrich Korea). Extracts were sequentially eluted with 250 μL of solvents of increasing polarity beginning with 100% ethyl acetate, followed by a mixture of ethyl acetate and acetonitrile (1:1, *v*/*v*), acetonitrile and methanol (1:1, *v*/*v*), and concluding with 100% methanol. The ethyl acetate fraction was then utilized to quantify EpOMEs. Each treatment was replicated three times with independent sample preparation.

### 2.5. LC-MS/MS Analyses

LC-MS/MS analysis was carried out using a QTrap 4500 (AB Sciex, Framingham, MA, USA) equipped with an auto-sampler, a binary pump, and a column oven. The analytical column was a C18 column (2.1 × 150 mm, 2.7 μm; Osaka Soda, Osaka, Japan) maintained at 40 °C. The mobile phases comprised 0.1% formic acid in water and 0.1% formic acid in acetonitrile. The gradient program started at 30% B at 0 min, maintained 30% B at 2 min, increased to 65% B by 12 min, further increased to 95% B by 12.5 min, remained at 95% B up to 25.0 min, and then reduced to 30% B at 28.0 min and held at 30% B until 30 min. The flow rate was set at 0.40 mL/min. The auto-sampler temperature was set at 5 °C and the injection volume was 10 μL. The LC-MS/MS was equipped with an electrospray ionization (ESI) source operating in negative ion mode. Post-optimization, the source parameters were temperature at 600 °C, curtain gas flow rate at 32 L/min, ion gas flow rate at 60 L/min, and the adjusted spray voltage at 4000 V. Analytical assessments were performed utilizing multiple reaction monitoring detection mode with nitrogen as the collision gas. Mass View1.1 software (AB Sciex) facilitated peak detection, integration, and quantitative analysis.

### 2.6. Bioinformatics to Predict EpOME Synthase (Fo-CYP) and Its Degradative Enzyme (Fo-sEH) Genes in F. occidentalis

To predict the EpOME synthase of *F. occidentalis*, all annotated *CYP* genes from *F. occidentalis*, previously archived in NCBI-GenBank (www.ncbi.nlm.nih.gov, accessed on 15 August 2024), were retrieved (Appendix A). These *CYP* genes were aligned with EpOME synthases from humans [35] and mice [36] and were also sourced from NCBI-GenBank (Appendix A). The predicted amino acid sequences of the candidate sEH genes (Appendix A) were compared with their orthologs across different species (Appendix A). Sequence alignments were conducted using Clustal W within MEGA 11 [37]. Phylogenetic trees for the CYPs and sEHs were generated using the neighbor-joining method, with 1000 replicates to calculate bootstrap values, employing MEGA 11. 

### 2.7. RNA Extraction, cDNA Synthesis, RT-PCR, and RT-qPCR

RNA was extracted from approximately 50 individuals at various developmental stages (larvae, pupae, adult males, and adult females) of *F. occidentalis* using Trizol reagent (Invitrogen, Carlsbad, CA, USA), following the manufacturer’s instructions. The extracted RNA was quantified using a spectrophotometer (NanoDrop, Thermo Fisher Scientific). For cDNA synthesis, 100 ng of RNA per reaction was used, employing RT-premix (Intron Biotechnology, Seoul, Republic of Korea) that included an oligo-dT primer, as per the manufacturer’s guidelines. The synthesized cDNA served as a template for PCR amplification with gene-specific primers (Appendix A). To assess cDNA integrity, the elongation factor 1 gene (*EF1*) was utilized. RT-PCR involved an initial heat treatment at 94 °C for 5 min followed by 35 cycles of 94 °C for 1 min for denaturation, an annealing temperature of 52 °C for 30 s, and an extension at 72 °C for 30 s, concluding with a final extension at 72 °C for 10 min. Gene expression levels were determined using a real-time PCR system (Step One Plus Real-Time PCR System, Applied Biosystems, Singapore) with Power SYBR Green PCR Master Mix (Toyobo, Osaka, Japan) following the manufacturer’s instructions. RT-qPCR was carried out in a 20 µL reaction volume containing 10 µL of 2× Power SYBR Green PCR Mix, 100 ng of cDNA template, and 10 pmol of each specific primer (Appendix A). The temperature cycles commenced with an initial heat treatment at 95 °C for 3 min, followed by 40 cycles of 98 °C for 20 s, 52 °C for 30 s, and 72 °C for 1 min. As an endogenous control, expression levels of *EF1* were analyzed in each sample. Single amplified products were confirmed by assessing the melting curves of the PCR products. Each treatment was replicated three times with independent samples. Expression analysis by qPCR was performed using the comparative Ct (2^−∆∆Ct^) method [38].

### 2.8. dsRNA Preparation and RNA Interference (RNAi)

Double-stranded RNAs (dsRNAs) were synthesized using a MEGAscript RNAi kit (Ambion, Austin, TX, USA), strictly following the manufacturer’s instructions. Initially, genes were amplified via PCR using gene-specific primers that incorporated a T7 RNA polymerase promoter sequence at the 5’ end (Appendix A). This PCR product served as the template for dsRNA synthesis. Both sense and antisense RNA strands were generated using T7 RNA polymerase at 37 °C over a period of 4 h. A control dsRNA (dsCON) was synthesized from a 520 bp segment of the viral gene CpBV302 [39]. The dsRNA was purified and mixed with the transfection agent Metafectene PRO (Biontex, Planegg, Germany) at a 1:1 (*v*/*v*) ratio and incubated at 25 °C for 30 min to facilitate liposome formation. Subsequently, the dsRNA–liposome complex was administered to larvae using a feeding protocol where beans were immersed in a 500 μg/mL dsRNA solution for 20 min. The efficiency of RNAi was assessed at intervals of 0, 6, 12, 24, and 48 h post-treatment through RT-qPCR, as previously described. Each experimental condition was replicated three times. 

### 2.9. Fluorescence In Situ Hybridization (FISH) Assay

After 12 h of feeding on the TSWV-inflicted diet, L1 larvae were dissected to extract the guts onto a sterilized slide glass, which were then fixed with 4% paraformaldehyde for 1 h at room temperature (RT). Following a rinse with 1× PBS, the guts were permeabilized with 1% Triton X-100 in PBS for 2 h at RT. Subsequent to another PBS wash, the guts were rinsed in 2× sodium saline citrate (SSC) and incubated at 42 °C with 25 µL of pre-hybridization buffer (2 µL of yeast tRNA, 2 µL of 20× SSC, 4 µL of dextran sulfate, 2.5 µL of 10% SDS, and 14.5 µL of deionized H_2_O) under dark and humid conditions for 1 h. The pre-hybridization buffer was then replaced with a hybridization buffer containing 5 µL of deionized formamide and 1 µL of fluorescein-labeled oligonucleotide in 19 µL of the initial pre-hybridization mix. DNA oligonucleotide probes targeting TSWV N (refer to Appendix A) were labeled at the 5′ end with fluorescein amidite (FAM) and subsequently purified using high-performance liquid chromatography (Bioneer, Daejeon, Republic of Korea). The slides were then covered with an RNase-free cover slip and maintained in a humid chamber at 42 °C overnight (18 h). Post-hybridization, the guts were washed twice with 4× SSC for 10 min each, followed by a 5 min incubation in 4× SSC containing 1% Triton X-100 at RT. After three washes with 4× SSC, the midgut samples underwent incubation at 37 °C in 1% anti-rabbit-FITC conjugated antibody (Thermo Fisher Scientific) in PBS in the dark for 30 min. The midgut was subsequently washed twice with 4× SSC for 10 min each and once with 3× SSC and was then allowed to air dry. After the application of a drop of 50% glycerol and a 15 min incubation at RT, the samples were covered by a cover glass and examined under a fluorescence microscope (DM2500, Leica, Wetzlar, Germany) at ×200 magnification.

### 2.10. Terminal Deoxynucleotidyl Transferase dUTP Nick End Labeling (TUNEL) Assay

TUNEL assays were conducted using the in situ Cell Death Detection kit (Abcam). Guts from L1 larvae were dissected in PBS on coverslips (22 × 22 mm). Subsequently, the tissues were exposed to 10 µM BrdU and terminal transferase for 1.5 h. Following fixation with 4% paraformaldehyde at RT for 1 h, the guts were rinsed with PBS and treated with 0.3% Triton-X in PBS for 2 h to permeabilize them. After a 1 h block with 5% BSA in PBS, the organs were incubated with anti-BrdU antibody (diluted 1:15 in the blocking solution) for 1 h at RT. Then, following three PBS washes to remove unbound anti-BrdU antibody, the organs were subjected to incubation with an FITC-conjugated anti-mouse IgG antibody (diluted 1:300 in blocking solution) for 1 h. After washing three times with PBS, 10 μL of DAPI (diluted 1:1000 in the blocking solution) was added and incubated at RT for 5 min. Finally, after rinsing the samples with PBS, a glycerol–PBS solution (10 μL) was applied to the sample on a coverslip, which was then mounted on a glass slide for observation under a fluorescence microscope (DFC450C, Leica) in FITC mode. Each treatment was replicated three times. 

### 2.11. Preparation of Recombinant pIB-NSs Vector

In a previous study [40], the recombinant pIB-NSs vector was constructed. Briefly, the NSs open reading frame (ORF: 1221 bp) was cloned into the pIB vector (pIB/V5-His TOPO TA Expression kit, Invitrogen) for in vivo transient expression (IVTE). After confirming the insertion direction by PCR, sequencing confirmed the accurate positioning of the ORF in the reading frame under the baculoviral immediate early promoter (OplE2). 

### 2.12. IVTE of NSs in a Nonhost, Spodoptera Exigua

Larvae were sourced from a laboratory strain of *S. exigua* and reared according to the method of Goh et al. [41]. IVTE was conducted using the protocol outlined by Hepat and Kim [42]. Briefly, the fourth instar larvae of *S. exigua* was injected with 1 μL of a mixture containing the pIB-NSs vector (200 ng) and a transfection reagent (Metafectene Pro, Biontex) at a 1:1 ratio using a Hamilton microsyringe. To evaluate expression levels, RNA was extracted from the experimental larvae at 0, 12, 24, 48, and 72 h post-injection (pi) using Trizol reagent (Invitrogen), following the manufacturer’s guidelines. qPCR was then performed to quantitatively analyze the gene expression, as previously described. 

### 2.13. Quantification of cAMP and Ca^2+^ Signals

cAMP measurement was conducted using a cyclic AMP ELISA kit (Cayman Chemical, Ann Arbor, MI, USA). The fourth instar larva of *S. exigua* was injected either with pIB-NSs vector (200 ng) alone or combined with dsRNA (500 ng) as previously described. At 24 h pi, 50 μL of hemolymph from the test larvae was combined with 148 μL of ACB, to which 2 μL of PGE_2_ (1 nM) was added. For the control group, the solvent for PGE_2_, namely, DMSO, was used in place of PGE_2_. This mixture was first centrifuged at 1000× *g* for 10 min at 4 °C. The resultant cell pellet was lysed in 100 μL of 0.1 M HCl for 20 min at RT, and the supernatant was collected following a second centrifugation at 1000× *g* for 10 min at 4 °C. The supernatant was then acetylated by adding 50 μL of 0.4 M KOH and acetic anhydride, both provided with the kit. After acetylation, 50 μL of each sample or standard was added to the designated wells along with sequential additions of 50 μL of cAMP AChE Tracer and 50 μL of cAMP EIA antiserum. The plate was incubated overnight at 4 °C. Following five washes with 200 μL of washing buffer, 200 μL of Ellman’s reagent was added, and the mixture was allowed to develop color in the dark at RT. Absorbance was measured at 405 nm. The cAMP concentration was determined by plotting an equation based on known standard concentrations, and the unknown concentration was subsequently calculated using this equation based on the absorbance values obtained at 405 nm.

To investigate the modulation of Ca^2+^ signals in hemocytes in response to the expression of the *NSs* gene, a combination of pIB-NSs vector (200 ng) and dsRNA (500 ng) was injected into fourth instar larvae of *S. exigua*. For the control group, dsCON was used, as mentioned earlier. After 24 h, the larvae were injected with 2 μL of Fura-8AM (1 mM) and PGE_2_ (1 μg/μL). At 1 h post-injection, hemolymph was collected and fixed on a slide glass using 2.5% paraformaldehyde. Cells exhibiting Fura fluorescence were observed under a fluorescence microscope at 200× magnification. Fluorescence intensity was analyzed using ImageJ (https://imagej.nih.gov/ij, accessed on 15 August 2024). 

### 2.14. Hemocyte-Spreading Behavior

To analyze the hemocyte-spreading behavior [29] in response to *NSs* expression, the vector and dsRNA were administered as described previously in the fourth instar larvae of *S. exigua*. After 24 h, total hemolymph (250 μL) was collected and combined with 750 μL of ACB. The hemocyte suspension was then chilled on ice for 30 min. Following centrifugation at 800× *g* for 5 min, 700 μL of the supernatant was removed, and the cell pellet was resuspended by adding 700 μL of TC-100 insect tissue culture medium (Welgene, Gyeongsan, Republic of Korea). The hemocyte suspension (19 μL) was then treated with 1 μL of PGE_2_ (1 μg/μL) on a glass coverslip for 1 h at RT. DMSO was used as the control instead of PGE_2_. Subsequently, cells were fixed with 4% paraformaldehyde for 10 min at RT. After three washes with PBS, cells were permeabilized with 0.2% Triton X-100 in PBS for 2 min at RT. Cells underwent a final wash in PBS and were then blocked with 10% BSA in PBS for 10 min at RT. Following another PBS wash, cells were incubated with FITC-tagged phalloidin in PBS for 1 h at RT. After three PBS washes, cells were treated with DAPI (1 mg/mL) in PBS for nuclear staining. Finally, after two additional washes in PBS, the cells were visualized under a fluorescence microscope (DM2500, Leica) at 200× magnification. Hemocyte spreading was assessed by measuring the extension of F-actin beyond the initial cell boundary. Each behavior assay was performed using 100 randomly selected cells. This procedure was repeated three times to ensure reproducibility of the results with independently prepared samples.

### 2.15. Nodulation Assay

The nodulation assay followed the method described by Vatanparast et al. [29]. Briefly, 24 h post-IVTE treatment, *E. coli* (2 ×10^4^ cells/larva) were injected into individual larvae of *S. exigua* and incubated for 8 h at RT. Subsequently, larvae were dissected to expose the hemocoel for counting melanized nodules using a stereoscopic microscope (Stemi SV11, Zeiss, Jena, Germany) at 50 × magnification. Each treatment was replicated three times.

### 2.16. Statistical Analysis

Continuous data were analyzed using one-way analysis of variance (ANOVA), executed in the PROC GLM module of the SAS [43]. Percent data were normalized through arcsine transformation. Differences among means were determined using the least significant difference (LSD) test at a Type I error rate of 0.05. All charts in this study were generated using GraphPad Prism v. 8.0.1 (Boston, MA, USA).

## 3. Results

### 3.1. Identification of EpOMEs in F. occidentalis

In terrestrial insects, including *F. occidentalis*, phospholipids are predominantly composed of linoleic acid (LA), which is typically released by the action of PLA_2_ and subsequently undergoes epoxidation by a specific CYP enzyme, leading to the production of 9,10- and 12,13-EpOMEs (Figure 1A). Following this, these EpOMEs are converted into 9,10- and 12,13-DiHOMEs by sEH. LC–MS/MS analysis of the L1 larvae of *F. occidentalis* revealed the presence of two EpOMEs and two DiHOMEs, which were further verified using tandem mass spectrometry based on their distinct ion peaks (Figure 1B and Appendix A). In the absence of viral infection, the larvae exhibited levels of 44.8 pg/larva for 9,10-EpOME and 40.59 pg/larva for 12,13-EpOME, with DiHOMEs being present at higher concentrations than EpOMEs. Nevertheless, these basal levels were significantly elevated due to TSWV infection, affecting both EpOME and DiHOME levels (Figure 1C).

### 3.2. EpOME Suppresses the Antiviral Response in F. occidentalis Infected with TSWV

Upon infection with TSWV, *F. occidentalis* larvae’s gut epithelium exhibited a robust positive signal of apoptosis from TUNEL assays (Figure 2A), with signals concentrated in the anterior midgut. However, the addition of 12,13-EpOME significantly reduced these apoptosis signals, similar to the impact of Z-VAD-FMK (ZVAD, a pan-caspase inhibitor), on the gut epithelium of *F. occidentalis*.

To explain the antiviral response based on the viral titers, TSWV was detected in the gut epithelium using FISH (Figure 2B) and quantified through RT-qPCR (Figure 2C). FISH analysis revealed TSWV throughout the intestine, including the Malpighian tubules of the infected larvae. Remarkably, the inclusion of arachidonic acid (AA, a biosynthetic precursor of eicosanoids) markedly reduced both the signal intensity and the viral titers. Additionally, AUDA (a specific inhibitor of sEH) was found to elevate the TSWV titer further. 

To further investigate the impact of EpOME on the antiviral response of *F. occidentalis*, apoptosis-associated genes (*Caspase 1* ~ *Caspase 4*) were evaluated for their expression levels (Figure 3). To elevate endogenous EpOME levels following the viral infection, AUDA was included with the TSWV infection. In comparison to the viral infection alone, the addition of AUDA resulted in significantly reduced expression levels of the four caspase genes. Conversely, the expression levels of the inhibitor of apoptosis (*IAP*) gene decreased post-viral infection, while the inclusion of AUDA mitigated the downregulation of *IAP* expression induced by the viral infection. Thus, AUDA addition suppressed caspase gene expressions while it prevented the decrease in the anti-apoptotic gene expression after the viral infection.

### 3.3. Prediction of EpOME Synthase in F. occidentalis and Its Functional Role in TSWV Infection

Utilizing the known EpOME synthase genes from mammals and *S. exigua* [27,35,36], the 18 designated CYPs of *F. occidentalis* were scrutinized to determine potential orthologs (Figure 4A). A neighbor-joining phylogenetic tree revealed that *Fo-CYP24* was grouped with the EpOME synthase genes. *Fo-CYP40* also demonstrated a close relationship with the EpOME synthase cluster. Both *Fo-CYP24* and *Fo-CYP40* were expressed throughout all developmental stages of *F. occidentalis* (Figure 4B). However, they exhibited distinct expression patterns, with *Fo-CYP24* showing high expression levels in the L1 stage, the pupal stage, and adult males, while *Fo-CYP40* was predominantly expressed during the L2 stage. Upon TSWV infection, *Fo-CYP24* expression was highly inducible, whereas *Fo-CYP40* expression was not (Figure 4C).

Individual RNAi treatments using dsRNAs specific to *Fo-CYP24* and *Fo-CYP40* significantly reduced the expression of the targeted genes over the subsequent 48 h pi, whereas the dsRNA control (dsCON) showed no difference (Figure 5A). The targeted silencing of *CYP24* expression resulted in a marked reduction in the TSWV titer in L1 larvae, whereas RNAi directed at *CYP40* did not affect the titer (Figure 5B). Further analyses of the TSWV titer in the intestine of the thrips via FISH (Figure 5C) revealed that in the dsCON treatment, the virus was detected throughout the intestine. Conversely, RNAi specific to *Fo-CYP24* expression significantly reduced the viral titer in the intestine, while RNAi directed at *Fo-CYP40* did not alter the viral titer.

### 3.4. Prediction of sEH of F. occidentalis and Their Functional Association with TSWV Infection

Four epoxide hydroxylase genes (*Fo-sEH1*, *Fo-sEH2*, *Fo-sEH3*, and *Fo-JHEH*) are encoded in the *F. occidentalis* genome. A phylogenetic tree analysis demonstrated that three EH genes clustered with other secretory EH (sEH) genes, while *Fo-JHEH* clustered with other microsomal EH (mEH) genes (Figure 6A). All three *Fo-sEH* genes and *Fo-JHEH* exhibited variations in their expression levels from L1 to adult stages (Figure 6B). *Fo-sEH1* and *Fo-sEH2* showed high expression levels in L1 larvae, whereas *Fo-sEH3* and *Fo-JHEH* were predominantly expressed in the adult stage. Upon TSWV infection, the expression level of *Fo-sEH2* decreased, whereas the expression levels of *Fo-sEH1*, *Fo-sEH3*, and *Fo-JHEH* genes remained unchanged (Figure 6C).

Individual RNAi treatments using dsRNA specific to the four *EH* genes significantly reduced their expressions during the subsequent 48 h pi (Figure 7A). Under these RNAi conditions, only the dsRNA specific to *Fo-sEH2* notably increased the TSWV titer at 24 h pi, while the other three RNAi treatments did not affect the control dsRNA treatment (Figure 7B). In observing the effects of RNAi in the intestine, TSWV was visualized by FISH (Figure 7C). The RNAi treatment with dsRNA specific to *Fo-sEH2* enhanced the viral titer in the intestine, whereas the other three did not.

### 3.5. A Virulence Factor of TSWV in F. occidentalis Involves an Increase in EpOME Levels

To determine the virulence factor(s) of TSWV responsible for the upregulation of host EpOME levels, three viral genes (*N*, *NSm*, and *NSs*) out of the five encoded in the TSWV genome were functionally assessed using a loss-of-function approach. Larvae infected with TSWV were treated with individual RNAi specific to *N*, *NSm*, or *NSs*, each targeted by their respective dsRNAs (dsRNA^N^, dsRNA^NSm^, and dsRNA^NSs^). These individual RNAi treatments effectively suppressed their target genes for at least 48 h pi (Figure 8A). Under these conditions, the expression levels of the EpOME synthase gene (*Fo-CYP24*) and the degradation gene (*Fo-sEH2*) were evaluated (Figure 8B). The RNAi treatment specific to *NSs* distinctly altered the expression of *Fo-CYP24*, unlike the other two treatments. Likewise, the RNAi specific to *NSs* also affected the expression of *Fo-sEH2*, in contrast to the other treatments. Therefore, RNAi targeting *NSs* prevented the upregulation of EpOME synthase gene expression while also failing to reduce the expression of the EpOME degradation gene in TSWV-infected larvae. To corroborate this regulatory effect of NSs on EpOME levels, the titers in virus-infected thrips were monitored following this specific RNAi treatment (Figure 8C). This RNAi intervention substantially suppressed the upregulation of EpOME titers across both regio-isomers, as determined by LC-MS/MS analysis.

### 3.6. TSWV-NSs Mimics the EpOME Action to the Immune Response of a Non-Host Insect, S. exigua

EpOME, serving as a negative regulator, plays a critical role in the immune responses of a lepidopteran insect, *S. exigua* [44]. To further investigate the role of NSs in EpOME-associated immune responses, its gene was cloned into a eukaryotic expression vector and injected into the larvae of *S. exigua* (Figure 9A). *NSs* transcript levels were highly detected at 48 h pi and subsequently decreased in *S. exigua*. With this in vivo transient expression (NSs-IVTE) of *NSs*, an immune inducer, PGE_2_, failed to induce hemocyte-spreading behavior, whereas larvae treated with the empty vector exhibited a significant upregulation of this behavior (Figure 9B). The inhibitory effect of NSs-IVTE was further confirmed through the addition of 12,13-EpOME to the PGE_2_ treatment. Notably, RNAi targeting NSs-IVTE with dsRNA^NSs^ abrogated the inhibitory effect of NSs on cellular behavior. This regulatory role of NSs in *S. exigua* was also demonstrated in a cellular immune response, assessed by nodule formation in response to a bacterial challenge. The influence of NSs on the hemocytes of *S. exigua* was further elucidated by monitoring the changes in secondary messenger levels following PGE_2_ exposure (Figure 9C). Treatment with PGE_2_ significantly elevated the internal levels of Ca^2+^ and cAMP. However, these secondary messenger levels were drastically reduced following treatments with EpOME or NSs-IVTE. Remarkably, the RNAi treatment specific to *NSs* expression significantly reversed the suppression of the secondary messenger signals.

## 4. Discussion

Although *F. occidentalis* is a competent vector of TSWV to a variety of host plants, this thrips species also mounts immune responses against the viral infection [45]. Eicosanoids are essential in mediating these antiviral responses in *F. occidentalis*, triggered by a damage signal from the midgut via DSP1 [20]. To enable efficient horizontal virus transmission, TSWV must suppress these antiviral responses and replicate within the insect vector. This study showed that TSWV modulates the levels of immunosuppressive EpOME in *F. occidentalis* to dampen the vector’s antiviral defenses.

This study utilized LC-MS/MS to identify EpOMEs and DiHOMEs in *F. occidentalis*. Previously, these C18 oxylipins were detected in only two other insect species: a lepidopteran, *S. exigua*, and a dipteran, *Culex quinquefasciatus* [29,46]. The presence of these compounds in thrips thus implies their presence in other insects as well, potentially playing a pivotal role in mediating physiological processes. The titers of EpOMEs increased in response to TSWV infection, suggesting that these oxylipins may play a role in the antiviral responses of *F. occidentalis* to TSWV. EpOMEs, derived from linoleic acid—a major component of insect phospholipids—are released through the catalytic activity of PLA_2_ [47]. Interestingly, PLA_2_ activity is significantly enhanced in *F. occidentalis* as early as 1 h after viral infection [45]. Additionally, our study identified *Fo-CYP24* as a potential EpOME synthase gene, with its expression also being inducible by the viral infection, albeit at a later stage. In *S. exigua*, EpOMEs are critical in curtailing excessive and unnecessary immune responses at the later stages of infection [27,28], which corresponds with the observed increase in EpOMEs in thrips following TSWV infection. This upregulation of EpOME levels may contribute to immune resolution in *F. occidentalis* in its response to TSWV. However, the onset of EpOME synthase induction in *F. occidentalis* occurred earlier (at 4 h post-infection) compared to that in *S. exigua* infected with a baculovirus (at 12 h post-infection), highlighting the potential differences influenced by the host manipulation of TSWV. This distinction can help elucidate the diverse interaction dynamics between different hosts and viruses.

Treatment of arachidonic acid, a precursor of eicosanoid biosynthesis, significantly reduced the TSWV titer in the virus-infected thrips. Eicosanoids mediate immune responses in insects [23]. They were pivotal in the antiviral responses of *F. occidentalis* against TSWV infection [45]. To investigate the role of EpOME in the immunosuppressive (antiviral) activity, an EpOME agonist, AUDA, was used because it is known to specifically inhibit sEH in other insects, such as mosquitoes and moths [27,48]. Moreover, this study identified *Fo-sEH2* as an EpOME-degrading enzyme, sEH. It appears that AUDA inhibited Fo-sEH2 to prevent the degradation of EpOMEs in *F. occidentalis*, thereby maintaining elevated endogenous levels of EpOMEs. AUDA treatment significantly increased the TSWV titer in the infected thrips, corroborating the role of EpOMEs in antiviral activity. To directly assess the EpOME activity, 12,13-EpOME was administered to the thrips. Notably, 12,13-EpOME was more effective than 9,10-EpOME in immunosuppression in *S. exigua* [28]. The introduction of 12,13-EpOME to the TSWV-infected thrips raised the viral titers by inhibiting apoptosis, similar to the effects observed with caspase inhibitor (ZVAD) treatment. Significantly, EpOME treatment suppressed the expression of various caspases in *F. occidentalis*, which were upregulated in response to TSWV. These findings underscore the role of EpOMEs in the antiviral response of *F. occidentalis* to TSWV infection. These results imply that maintaining relatively high levels of EpOME in the thrips vector could benefit TSWV multiplication and its horizontal transmission. 

It was subsequently queried which viral factor(s) influence the EpOME levels in *F. occidentalis*. Among the five genes encoded in the TSWV genome, three genes—*NSs*, *NSm*, and *N*—were selectively tested by individual RNAi treatments, as the remaining two genes, *RdRp* and *Gn/Gc*, are recognized, respectively, as genome replication and viral coat proteins [49]. RNAi treatment targeting *NSs* expression notably failed to induce *Fo-CYP24* expression and to suppress *Fo-sEH2* expression, whereas the other two RNAi treatments did not exhibit these effects. The encoded product of *NSs* found in the S segment of TSWV is critical for the horizontal transmission of TSWV among plant hosts via the thrips vector. The importance of the *NSs* product in viral transmission was evidenced by experiments using *NSs*-truncated TSWV isolates, which were unable to propagate viral particles within the vector thrips [50]. The physiological significance of the *NSs* product is suggested by its role in suppressing host RNAi efficiency, which helps prevent the degradation of the viral genome by the host’s RISC activity [51]. Orthologous gene products, similar to NSs, are present in various Bunyaviruses infecting both plants and vertebrates [52]. Notably, these products commonly bind both small and long dsRNAs in tospoviruses, acting as RNA silencing suppressors against host plant immune defenses [53]. The N-terminal domain of NSs is pivotal for the RNA silencing suppression function in host plants [54]. Additionally, site-directed mutagenesis assays identified two conserved motifs, GKV/T at positions 181–183 and YL at positions 412–413, as critical for the silencing suppressor function of NSs [55]. Interestingly, even in the non-host insect, *S. exigua*, the NSs protein suppresses RNAi activity, which renders it highly susceptible to iflaviral infections [40]. This suggests that NSs might manipulate the expression of genes, like *Fo-CYP24* and *Fo-sEH2*, through its silencing suppressive impact on unknown regulatory RNA at a post-transcriptional level, or it could directly affect their gene expressions. Further research is required to clarify these roles.

Altogether, our current study showed that EpOMEs act as immune resolving-like molecules in *F. occidentalis*. TSWV upregulates EpOME biosynthesis to suppress the antiviral responses of the thrips through its NSs protein. This study proposes a novel molecular interaction between TSWV and its insect vector for viral transmission. Notably, NSs suppress host antiviral responses through two distinct molecular activities: RNA silencing suppression and EpOME biosynthesis induction.

## Figures and Tables

**Figure 1 cells-13-01377-f001:**
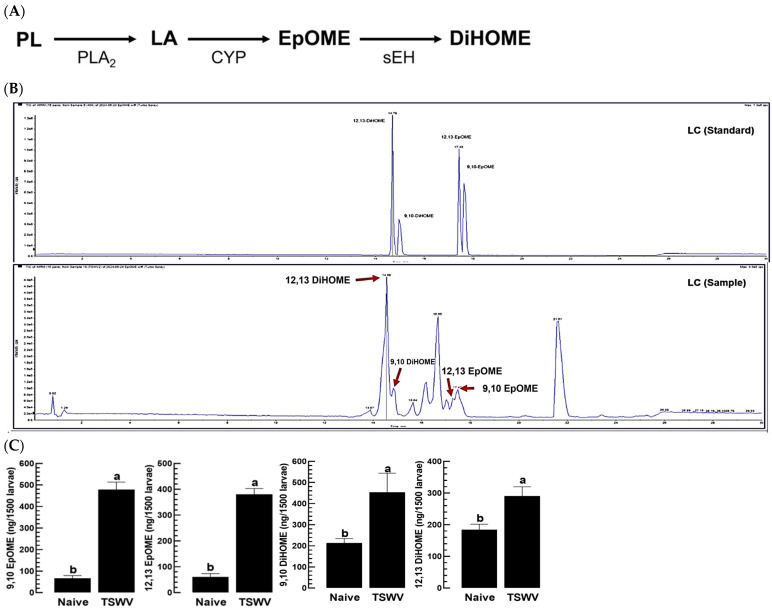
Detection of EpOMEs in TSWV-infected and naïve *F. occidentalis* larvae. (**A**) A hypothesized pathway of EpOME biosynthesis in *F. occidentalis*. Phospholipase A_2_ (‘PLA_2_’) catalyzes phospholipid (‘PL’) to release linoleic acid (‘LA’), which is then oxidized by cytochrome P450 monooxygenase (‘CYP’) to yield EpOME. The epoxy ring of EpOME is hydrolyzed by soluble epoxide hydrolase (‘sEH’) to produce DiHOME. (**B**) Chromatograms of LC–MS/MS for detecting EpOMEs extracted from L1 larvae of *F. occidentalis*. (**C**) Comparative assessments of EpOME levels in TSWV-infected and naïve *F. occidentalis* larvae. After 24 h of TSWV exposure, 1500 L1 larvae from each group were analyzed for EpOME levels using LC–MS/MS. The analysis was conducted in triplicate for each sample, with significant differences in means designated by different letters above the standard error bars (Type I error = 0.05, LSD test).

**Figure 2 cells-13-01377-f002:**
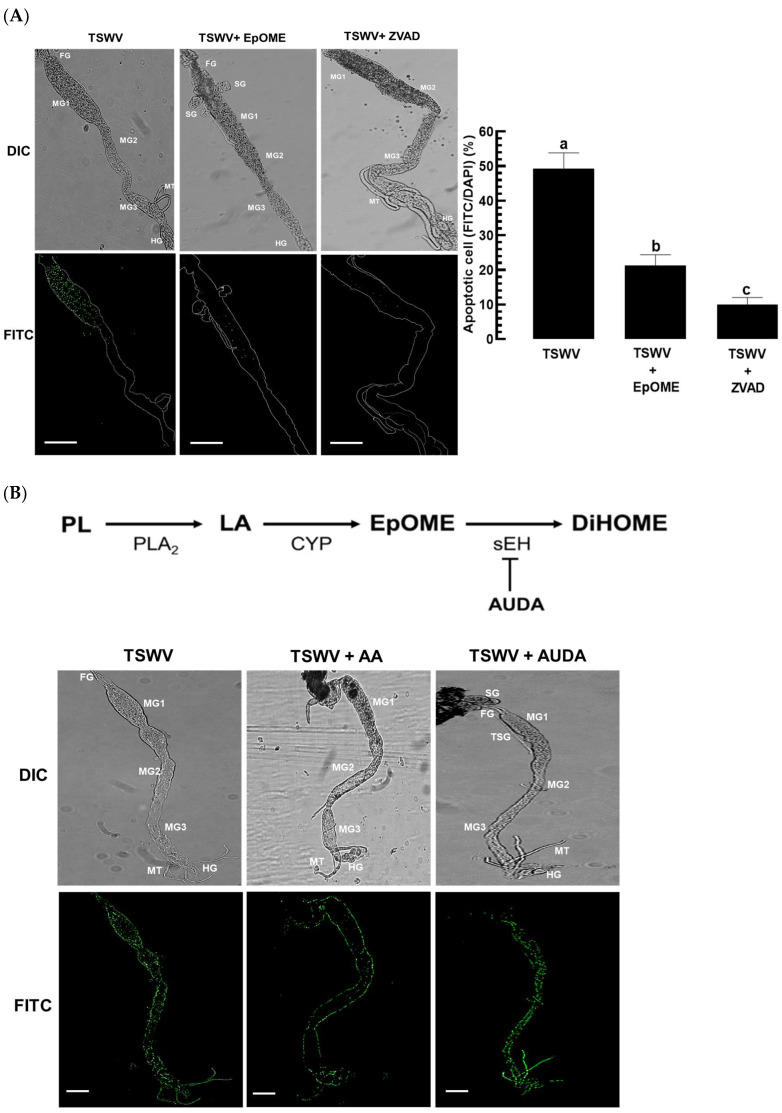
Suppressive effect of EpOME on the antiviral response of *F. occidentalis* against TSWV infection. (**A**) Apoptosis in the gut epithelium of *F. occidentalis* at 12 h after TSWV infection, evaluated by TUNEL assay. 12,13-EpOME (0.1 μg/mL) or Z-VAD-FMK (‘ZVAD’, 50 μM) was introduced during the viral infection. An anti-mouse IgG antibody specific to BrdU confirmed a positive TUNEL response under a fluorescent microscope (DM2500; Leica, Wetzlar, Germany). The whole tissues and the apoptotic cells were observed by differential interference contrast (‘DIC’) and fluorescein isocyanate (‘FITC’) modes, respectively. Both FITC and DAPI signals were quantified in 100 randomly selected cells across three replicates. White lines in FITC mode delineate the contour of the internal organs of the thrips. (**B**) Impact of arachidonic acid (‘AA’, 1 µg/mL) or AUDA (10 ppm) on the viral titers within the guts, evaluated via the FISH assay. An anti-sense probe specific to the *TSWV-N* gene (Appendix A) was utilized. The corresponding sense probes did not yield any detectable signal (Appendix A). (**C**) Relative TSWV titers throughout the insects were measured by RT-qPCR at 12 h post-viral treatments. Different letters atop the standard error bars signify significant differences among means at a Type I error rate of 0.05 (LSD test). The scale bars denote 0.1 mm.

**Figure 3 cells-13-01377-f003:**
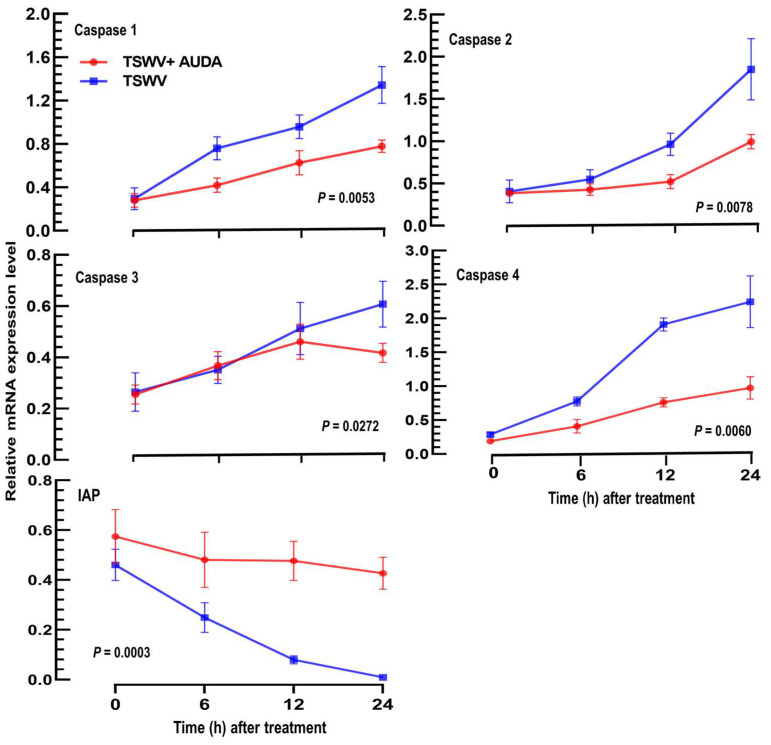
Suppressive effect of EpOME on the caspase gene expressions induced by TSWV infection in *F. occidentalis*. To elevate the endogenous levels of EpOME, its degradation was suppressed using AUDA, a specific suppressor of sEH. AUDA (10 ppm) was administered alongside the viral treatment. The expression levels of four caspase genes, along with an inhibitor of the apoptosis (*IAP*) gene, were monitored following the viral treatments using RT-qPCR. Each measurement was replicated three times. The *p* values provided indicate the Type I error between two treatments at four different time points.

**Figure 4 cells-13-01377-f004:**
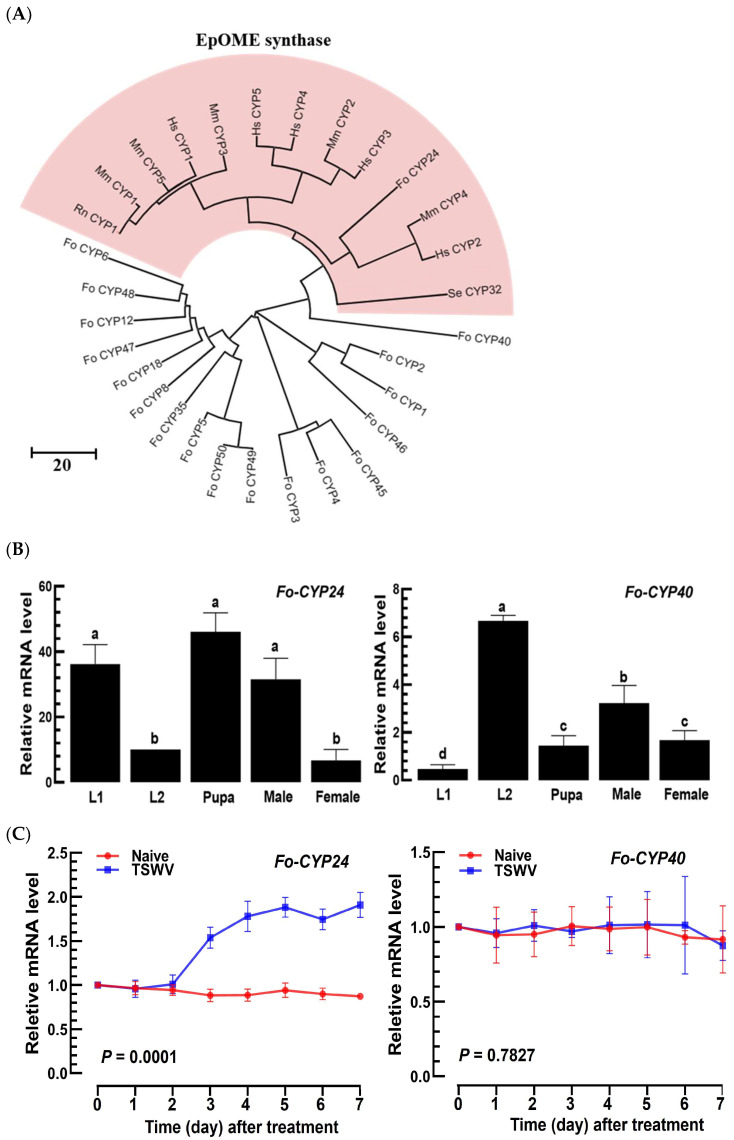
Prediction of EpOME synthase in *F. occidentalis*. (**A**) Phylogenetic analysis of *F. occidentalis CYP* genes was performed alongside EpOME synthase genes of mammals and *S. exigua* based on their amino acid sequences (refer to GenBank accession numbers in Appendix A). Neighbor-joining clustering employed MEGA 11 software, with bootstrapping values from 1000 repetitions providing support for branching and clustering. (**B**) Expression patterns of *Fo-CYP24* and *Fo-CYP40* were assessed at different developmental stages of *F. occidentalis*. Distinct letters above the standard error bars denote significant differences in means at a Type I error rate of 0.05 (LSD test). (**C**) The relative mRNA levels of *Fo-CYP40* and *Fo-CYP24* following a TSWV challenge in *F. occidentalis*, determined via RT-PCR. Each measurement was replicated three times. *p* values indicated the Type I error between two treatments at four distinct time points.

**Figure 5 cells-13-01377-f005:**
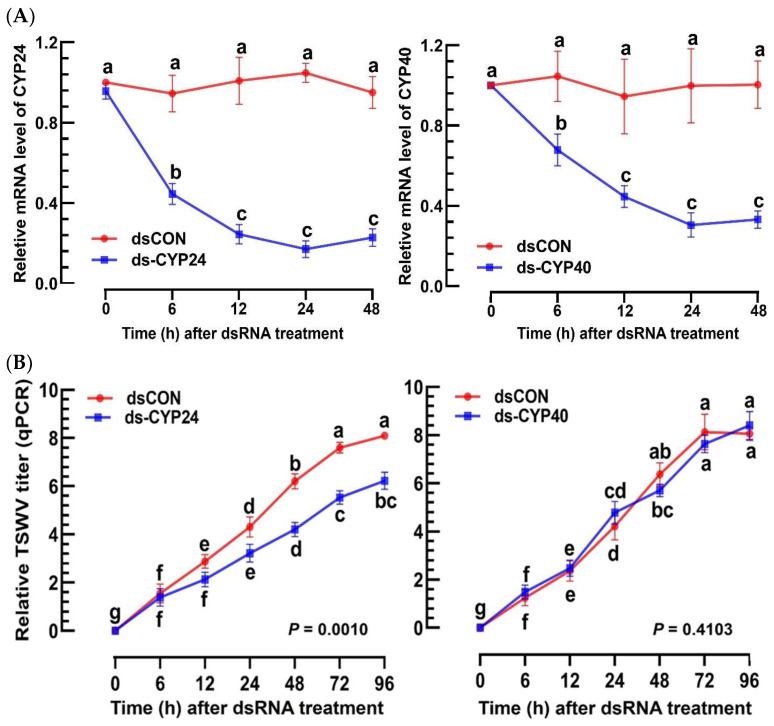
Functional assay of EpOME synthase candidate genes in *F. occidentalis* by measuring the TSWV titers. (**A**) RNAi treatments targeting *Fo-CYP24* or *Fo-CYP40* expressions in L1 larvae involved feeding the larvae with specific dsRNAs (‘dsCYP24’ and ‘dsCYP40’) at 500 µg/mL. A non-target gene, *CpBV302*, served as a control dsRNA (dsCON). (**B**) Changes in the relative TSWV titers following the RNAi treatments were assessed 12 h post-treatment using RT-PCR. (**C**) Changes in TSWV titers in the intestines of thrips post-RNAi treatment were observed using FISH. The tissues and virus were visualized using differential interference contrast (‘DIC’) and fluorescein isocyanate (‘FITC’), respectively. Both FITC and DAPI signal intensities were quantified in 100 randomly selected cells over three replicates. The scale bars represent 0.1 mm. Each treatment was replicated three times. Different letters above standard error bars indicate significant differences among means at Type I error = 0.05 (LSD test).

**Figure 6 cells-13-01377-f006:**
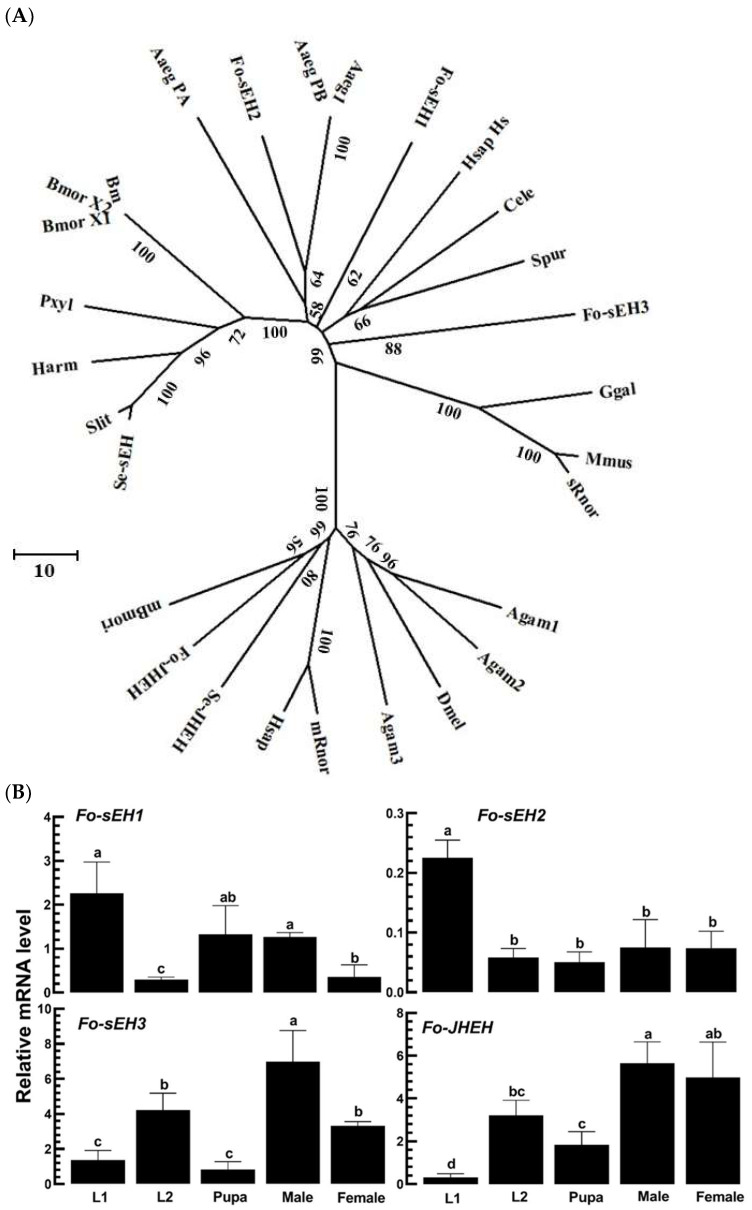
Prediction of an EpOME-degrading enzyme, *sEH*, gene in *F. occidentalis*. (**A**) Phylogenetic analysis of *F. occidentalis sEH* genes with other insect sEHs in their amino acid sequences (refer to their GenBank accession numbers in Appendix A). Neighbor-joining clustering was conducted using MEGA 11. Bootstrapping with 1000 repetitions was employed to confirm the support for branching and clustering. (**B**) Expression patterns of three *sEH* and one juvenile hormone epoxide hydroxylase (*JHEH*) genes of *F. occidentalis* across various developmental stages. Distinct letters above the standard error bars denote significant differences among the means at a Type I error rate of 0.05 (LSD test). (**C**) Relative mRNA levels of four *EH* genes in *F. occidentalis* following a TSWV challenge by RT-PCR. Each assay was replicated three times. *p* values denote the Type I error between two treatments at four distinct time points.

**Figure 7 cells-13-01377-f007:**
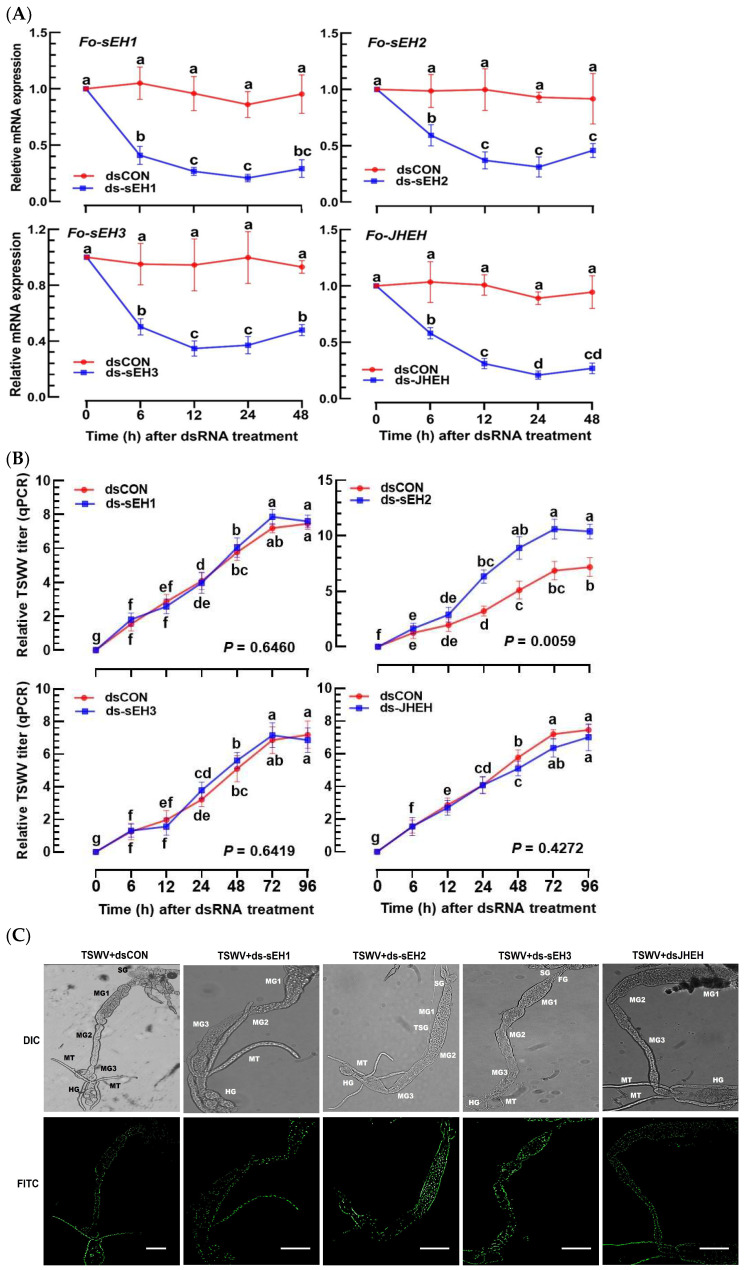
Functional assay of *sEH* candidate genes in *F. occidentalis* by measuring the TSWV titers. (**A**) Individual RNAi treatments targeting three sEH and JHEH gene expressions in L1 larvae were conducted. The L1 larvae received specific dsRNAs (‘ds-sEH1–ds-sEH3’ and ‘dsJHEH’) at a concentration of 500 µg/mL. A non-target gene, *CpBV302*, served as the control dsRNA (dsCON). (**B**) Assessment of changes in relative TSWV titers following the RNAi treatments. TSWV was introduced 12 h post-RNAi treatment, and its titers were measured via RT-PCR. (**C**) Examination of the changes in TSWV titers within the thrips’ intestines post-RNAi treatment using FISH. Tissues and the virus were visualized using differential interference contrast (‘DIC’) and fluorescein isocyanate (‘FITC’) techniques. Intensities of both FITC and DAPI signals were quantified from 100 randomly selected cells across three replicates. The scale bars denote 0.1 mm. Each experimental treatment was replicated thrice. Different letters above the standard error bars signify statistically significant differences among the means at a Type I error rate of 0.05 (LSD test).

**Figure 8 cells-13-01377-f008:**
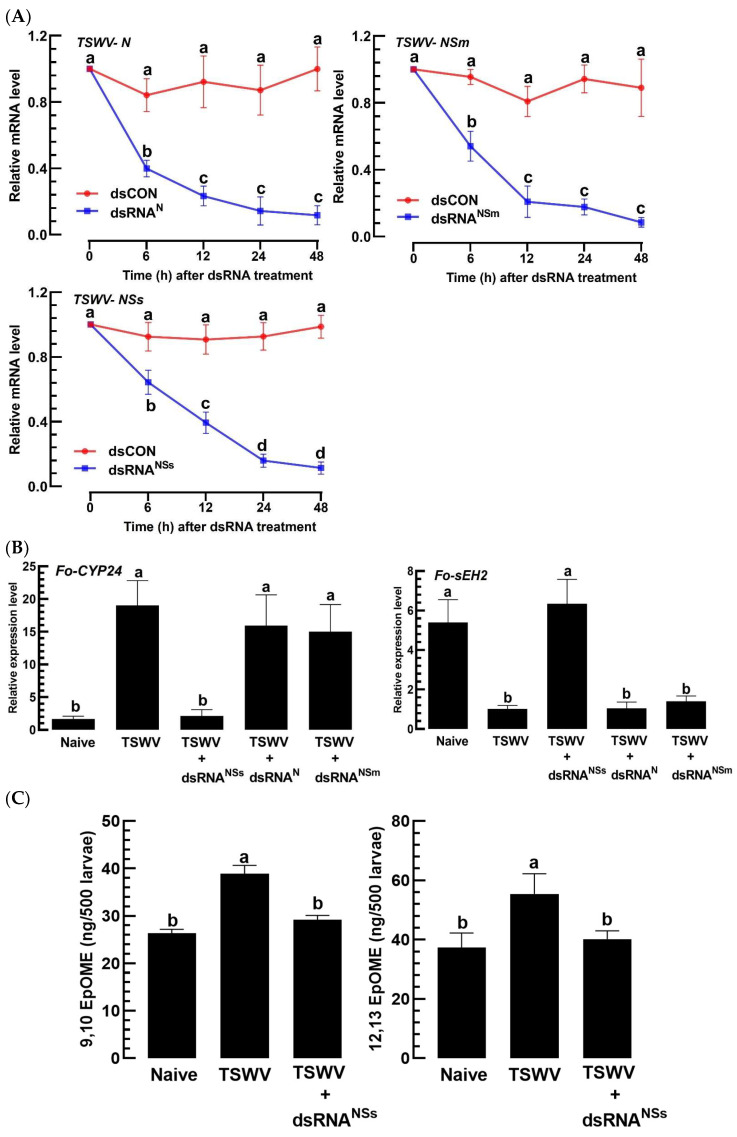
NSs function as a virulence factor of TSWV, promoting the elevation of EpOME levels to dampen the antiviral responses in *F. occidentalis*. In panel (**A**), RNAi treatments targeted at three TSWV gene expressions (*N*, *NSm*, and *NSs*) were administered by feeding the gene-specific dsRNAs (dsRNA^N^, dsRNA^NSm^, and dsRNA^NSs^) at a dose of 500 µg/mL to L1 larvae. A non-target gene, *CpBV302*, served as a control dsRNA (dsCON). Panel (**B**) details the effects of these RNAi treatments on the expression levels of *Fo-sEH2* and *Fo-CYP24*. At 12 h post-RNAi treatment, TSWV was administered to L1 larvae, and the expression levels of *Fo-sEH2* and *Fo-CYP24* were assessed by RT-PCR. Panel (**C**) reveals that RNAi specifically targeting NSs expression interferes with the upregulation of EpOME levels induced by TSWV in *F. occidentalis*. Each treatment was replicated three times. Distinct letters above the standard error bars signify significant differences among the means at a Type I error rate of 0.05 (LSD test).

**Figure 9 cells-13-01377-f009:**
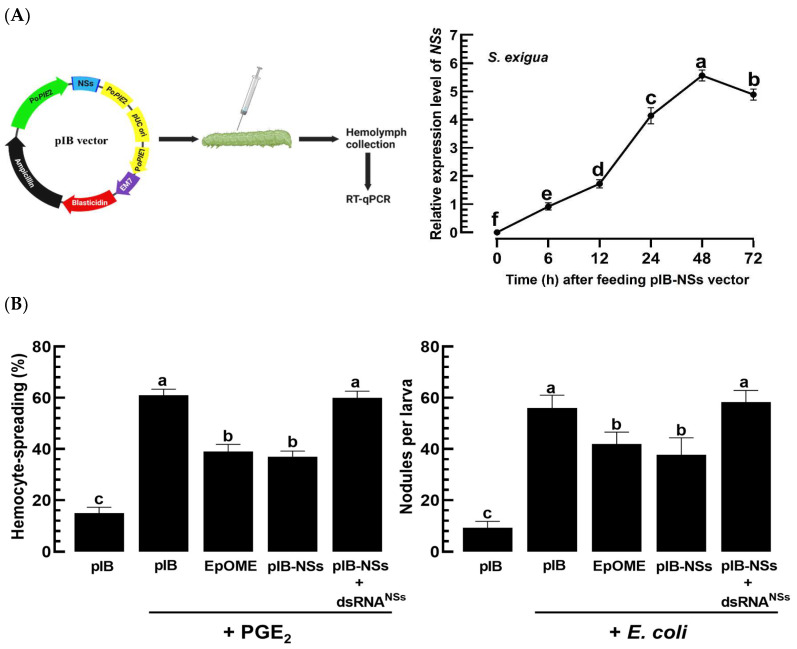
In vivo transient expression (IVTE) of *NSs* gene of TSWV and its suppression of immune responses of *S. exigua*. (**A**) IVTE of NSs in *S. exigua* involved injecting the expression vector (200 ng/larva) into L4 larvae. (**B**) Effects of NSs-IVTE on cellular immune responses included hemocyte-spreading behavior and nodule formation. For hemocyte spreading, cells were incubated with 1 μL of PGE_2_ (1 μg/μL) for 40 min, and spreading behavior was quantified by counting the cells showing F-actin growth (stained by FITC) beyond cell boundaries among 100 randomly chosen cells. The number of nodules was counted 8 h after bacterial infection (2 × 10^4^ cells/larva). (**C**) The impact of NSs-IVTE on Ca^2+^ and cAMP levels in hemocytes was also examined. L4 larvae received injections of either the pIB-NSs vector (200 ng) or the pIB-NSs vector + dsRNA^NSs^ (500 ng). Controls were injected with empty pIB (200 ng) and 12,13-EpOME (0.1 µg/larva). Ca^2+^ signals were quantified based on the intensities in FITC, normalized with the number of the observed cells. Scale bars indicate 10 μm. Each experiment was replicated three times. Statistical significance was determined using the LSD test at Type I error = 0.05, with different letters above the standard error bars denoting significant differences among means.

## Data Availability

The original contributions presented in the study are included in the article/Appendix A, further inquiries can be directed to the corresponding author.

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
