# Peer review of "Tomato Spotted Wilt Virus Suppresses the Antiviral Response of the Insect Vector, Frankliniella occidentalis, by Elevating an Immunosuppressive C18 Oxylipin Level Using Its Virulent Factor, NSs"

_cells, 2024, doi:10.3390/cells13161377_

Round 1

Reviewer 1 Report

Comments and Suggestions for Authors

The tomato spotted wilt virus (TSWV), Orthotospovirus tomatomaculae, is a significant economic threat to global agriculture due to its ability to infect more than 1000 plant species spanning over 90 families. In this study, the authors hypothesize that TSWV upregulates the EpOME signal that antagonizes induced immune responses in F. occidentalis. The authors were able to show that EpOMEs act as immune resolving-like molecules in F. occidentalis. TSWV up-regulates EpOME biosynthesis to suppress the antiviral responses of the thrips through its NSs protein. The article is well and concisely written and shows interesting results and data on a novel molecular interaction between TSWV and its insect vector for viral transmission. I recommend accepting the article for publication after clarifying a few points:

1. The materials and methods section is well and thoroughly described. However, there are no references to almost all the methods used. The authors should carefully review this section and insert references to the methods they used.

2. Lines 214-215 should specify in the text of the article which comparative CT method was used (2-ΔΔCT )

3. Section 3.2. The paragraph on lines 431-438 discusses the expression of four caspase genes. I would like to see a little more information about them, except that they are associated with apoptosis.

Author Response

Comment #1-1: The materials and methods section is well and thoroughly described. However, there are no references to almost all the methods used. The authors should carefully review this section and insert references to the methods they used.

Response: Appropriate references are added as follows:

2.3. TSWV infection to thrips

TSWV was isolated from hot pepper leaves exhibiting typical viral symptoms, in-cluding ring spots, in Andong, Korea, and verified with an Immunostrip TSWV kit (Agdia, Elkhart, IN, USA). Approximately 100 mg of the plant tissues were homoge-nized in 1 mL of filter (0.22 µm pore size)-sterilized PBS and centrifuged at 14,000 × g for 5 min. The supernatant was employed as the virus suspension. The viral infection followed the method described by Kim et al. [20]. Briefly, prior to the immune chal-lenge, L1 or L2 larvae…….

2.4. Insect sample preparation to quantify EpOMEs

L1 larvae of F. occidentalis were utilized for the extraction of EpOMEs. Following a 24-hour diet that was infected with TSWV, 1,500 larvae were gathered and rinsed three times with chilled PBS. These sample preparations were independently replicated three times. EpOME extraction followed the method described by Vatanparast et al. [2]. Briefly, each sample underwent three cycles of homogenization…….

2.14. Hemocyte-spreading behavior

To analyze the hemocyte-spreading behavior [29] in response to NSs expression, the…….

2.15. Nodulation assay

Nodulation assay followed the method described by Vatanparast et al. [29]. Briefly, 24 h post-IVTE treatment,

Comment #1-2: Lines 214-215 should specify in the text of the article which comparative CT method was used (2-ΔΔCT )

Response: added

Comment #1-3: Section 3.2. The paragraph on lines 431-438 discusses the expression of four caspase genes. I would like to see a little more information about them, except that they are associated with apoptosis.

Response: Necessary information is added as follows:

“To further investigate the impact of EpOME on the antiviral response of F. occidentalis, apoptosis-associated genes (Caspase 1 ~ Caspase 4) were evaluated for their expression levels (Figure 3). To elevate endogenous EpOME levels following the viral infection, AUDA was included with the TSWV infection. In comparison to the viral infection alone, the addition of AUDA resulted in significantly reduced expression levels of the four caspase genes. Conversely, the expression levels of the inhibitor of apoptosis (IAP) gene decreased post-viral infection, while the inclusion of AUDA mitigated the down-regulation of IAP expression induced by the viral infection. Thus, AUDA addition suppressed caspase gene expressions while it prevented the decrease of the an-ti-apoptotic gene expression after the viral infection.”

Reviewer 2 Report

Comments and Suggestions for Authors

The manuscript by Shahmohammadi et al. investigates the mechanisms by which tomato spotted wilt virus (TSWV) modulates the immune response of its insect vector, Frankliniella occidentalis. Given the economic impact of TSWV on crop production, understanding these interactions is critical for developing effective control strategies. The manuscript is well-written and well-organized, presenting the research findings in a clear and logical manner. However, several aspects require attention and correction to improve its overall quality and clarity.

1. The abstract is informative but dense. Simplifying sentences and removing redundant phrases could enhance readability. The journal recommends a length of around 200 words, but the current abstract exceeds 300 words. Condense the content to meet the word limit and maintain essential information.

2. Lines 101-103: Repetition of the phrase “This study” in consecutive sentences (last paragraph of the Introduction section) should be considered for rephrasing to avoid redundancy. For example: “This study proposed a hypothesis that TSWV upregulates the EpOME signal, which antagonizes the induced immune responses in F. occidentalis. To evaluate this hypothesis, we monitored the EpOME levels in thrip post-viral infection and ….”.

3. Lines 111: Omit “and” in the sentence “A laboratory population of F. occidentalis originated from hot pepper (Capsicum annuum L.) fields in Andong, Korea and was cultured at a temperature of….”

4. Line 170, 172: Ensure consistent use of units and formatting (e.g., “min” instead of “mins”; “auto-sampler” instead of “auto -sampler”).

5. Lines 174-175: One instance of “source” can be removed to avoid redundancy.

6. Lines 186-187: Replace “The candidate sEH genes predicted amino acid sequences” with “The predicted amino acid sequences of the candidate sEH genes” for better readability.

7. Line 210: Using “an initial heat treatment at 95°C for 10 min” for cDNA is unusual since this step is typically for denaturing double-stranded DNA in PCR. For single-stranded cDNA, a shorter initial denaturation is generally sufficient.

8. Line 215: “CT method” should be corrected to “Ct method” for consistency with standard terminology.

9. Lines 223-227 - Simplify the sentences “Additionally, a control dsRNA (dsCON), consisting of a 520 bp dsRNA segment from CpBV302, a viral gene [39], was synthesized. The synthesized dsRNA was then purified, combined with the transfection agent Metafectene PRO (Biontex, Planegg, Germany) ….” to “A control dsRNA (dsCON) was synthesized from a 520 bp segment of the viral gene CpBV302 [39]. The dsRNA was purified and mixed with the transfection agent Metafectene PRO (Biontex, Planegg, Germany)…”.

10. The authors mention using differential interference contrast (DIC) microscopy, but they did not specify the microscopy technique and microscope model in the Materials and Methods section.

In general, the manuscript is solid and provides valuable insights into TSWV interaction with its insect vector, but some parts would benefit from minor revisions to improve clarity, conciseness and organisation.

Comments on the Quality of English Language

The manuscript is generally well-written and organised, but requires minor grammatical corrections and simplification of certain sections to improve readability.

Author Response

Comment #2-1: The abstract is informative but dense. Simplifying sentences and removing redundant phrases could enhance readability. The journal recommends a length of around 200 words, but the current abstract exceeds 300 words. Condense the content to meet the word limit and maintain essential information.

Response: The abstract size is now reduced into 229 words from over 300 words.

Comment #2-2: Lines 101-103: Repetition of the phrase “This study” in consecutive sentences (last paragraph of the Introduction section) should be considered for rephrasing to avoid redundancy. For example: “This study proposed a hypothesis that TSWV upregulates the EpOME signal, which antagonizes the induced immune responses in F. occidentalis. To evaluate this hypothesis, we monitored the EpOME levels in thrip post-viral infection and ….”.

Response: Replaced as suggested

Comment #2-3: Lines 111: Omit “and” in the sentence “A laboratory population of F. occidentalis originated from hot pepper (Capsicum annuum L.) fields in Andong, Korea and was cultured at a temperature of….”

Response: Deleted

Comment #2-4: Line 170, 172: Ensure consistent use of units and formatting (e.g., “min” instead of “mins”; “auto-sampler” instead of “auto -sampler”).

Response: Corrected

Comment #2-5: Lines 174-175: One instance of “source” can be removed to avoid redundancy.

Response: Removed as follows: “Post-optimization, the source parameters were temperature at 600â—¦C, curtain gas flow rate at 32 L/min, ion gas flow rate at 60 L/min, and the adjusted spray voltage at 4,000 V.”

Comment #2-6: Lines 186-187: Replace “The candidate sEH genes predicted amino acid sequences” with “The predicted amino acid sequences of the candidate sEH genes” for better readability.

Response: Replaced

Comment #2-7: Line 210: Using “an initial heat treatment at 95°C for 10 min” for cDNA is unusual since this step is typically for denaturing double-stranded DNA in PCR. For single-stranded cDNA, a shorter initial denaturation is generally sufficient.

Response: Corrected into 3 min

Comment #2-8: Line 215: “CT method” should be corrected to “Ct method” for consistency with standard terminology.

Response: Corrected

Comment #2-9: Lines 223-227 - Simplify the sentences “Additionally, a control dsRNA (dsCON), consisting of a 520 bp dsRNA segment from CpBV302, a viral gene [39], was synthesized. The synthesized dsRNA was then purified, combined with the transfection agent Metafectene PRO (Biontex, Planegg, Germany) ….” to “A control dsRNA (dsCON) was synthesized from a 520 bp segment of the viral gene CpBV302 [39]. The dsRNA was purified and mixed with the transfection agent Metafectene PRO (Biontex, Planegg, Germany)…”.

Response: Replaced as recommended

Comment #2-10: The authors mention using differential interference contrast (DIC) microscopy, but they did not specify the microscopy technique and microscope model in the Materials and Methods section.

Response: The information in Figure 9 caption is deleted. The sentence is rephrased as follows: “Ca2+ signals were quantified based on the intensities in FITC, normalized with the number of the observed cells.”  
